# Immune Checkpoint Blockade Enhances Immune Activity of Therapeutic Lung Cancer Vaccine

**DOI:** 10.3390/vaccines8040655

**Published:** 2020-11-05

**Authors:** Pournima Kadam, Ram P. Singh, Michael Davoodi, Jay M. Lee, Maie St. John, Sherven Sharma

**Affiliations:** 1Department of Medicine, Veterans Affairs Greater Los Angeles Healthcare System, Los Angeles, CA 90073, USA; Pournima.kadam2@va.gov (P.K.); Ram.Singh@va.gov (R.P.S.); michael.davoodi@ucla.edu (M.D.); 2Department of Medicine, UCLA Lung Cancer Research Program, David Geffen School of Medicine at UCLA, Los Angeles, CA 90095, USA; JaymoonLee@mednet.ucla.edu; 3Jonsson Comprehensive Cancer Center, David Geffen School of Medicine at UCLA, Los Angeles, CA 90095, USA; MStJohn@mednet.ucla.edu; 4Department of Head and Neck Surgery, David Geffen School of Medicine at UCLA, Los Angeles, CA 90095, USA; 5UCLA Head and Neck Cancer Program, David Geffen School of Medicine at UCLA, Los Angeles, CA 90095, USA

**Keywords:** lung cancer, PD-1, CCL21, dendritic cells (DC), T cells, tumor microenvironment (TME)

## Abstract

Background: Immune checkpoint blockade that downregulates T cell evasion for effective immunity has provided a renewed interest in therapeutic cancer vaccines. Methods: Utilizing murine lung cancer models, we determined: tumor burden, TIL cytolysis, immunohistochemistry, flow cytometry, RNA Sequencing, CD4 T cells, CD8 T cells, CXCL9 chemokine, and CXCL10 chemokine neutralization to evaluate the efficacy of Programmed cell death protein 1 (PD-1) blockade combined with chemokine (C-C motif) ligand 21-dendritic cell tumor antigen (CCL21-DC tumor Ag) vaccine. Results: Anti-PD1 combined with CCL21-DC tumor Ag vaccine eradicated 75% of 12-day established tumors (150 mm^3^) that was enhanced to 90% by administering CCL21-DC tumor Ag vaccine prior to combined therapy. The effect of combined therapy was blocked by CD4, CD8, CXCL9, and CXCL10 neutralizing antibodies. Conclusion: PD-1 blockade therapy plus CCL21-DC tumor Ag vaccine could be beneficial to lung cancer patients.

## 1. Introduction

Lung cancer is the leading cause of cancer death worldwide [1]. The long-term survival after resection for non-small cell lung cancer (NSCLC) is about 50%. Novel effective therapeutic strategies are needed for long-term survival outcome. With the approval of ipilimumab, PD-1, and PD-L1 inhibitors for NSCLC, immune oncology has revolutionized cancer therapy, though not all patients respond [2]. Combination immunotherapy has the potential for allowing long-term cancer-free survival by targeting the immune activating and evasion pathways. This can be achieved with therapeutic vaccines and immune checkpoint blockade to increase the activated T cell response in the tumor microenvironment (TME). PD-L1 and PD-L2 ligands inhibit T cell immune responses by interacting with PD1 receptors and conferring tolerance. In a recent study, we demonstrated that CCL21-DC tumor lysate vaccine combined with anti-PD1 blockade led to tumor eradication in the Kras-G12Dp53null tumor model [3]. In this study, we are evaluating the generality of this observation by combining DC with tumor Ag and CCL21 to enhance T cell infiltration of tumors. CCL21 chemokine recruits Ag-loaded DC into T cell areas through chemokine receptor 7(CCR7) and C-X-C motif chemokine receptor 3 (CXCR3) that culminate in cognate T cell activation [3]. The advantage of s.c. vaccine is the ability of non-invasive repeat dosing to elicit T cell-dependent, systemic anti-tumor responses. The purpose of this study is to understand the mechanisms of PD-1 blockade combined with therapeutic CCL21-DC tumor Ag vaccination in lung cancer. For patients who have reduced CD8 T cells in their tumors and/or low expression of tumor PD-L1, CCL21-DC tumor Ag vaccine will be of clinical benefit.

## 2. Material &Methods

### 2.1. Cell Line and Reagents Utility

The LKR-13 K-RasG12D lung adenocarcinoma cell line was obtained from J. Kurie (MD Anderson). The reagents obtained, used and cells were cultured in RPMI 1640 as previously described [3]. Immunofluorescence (IF) Ab for perforin and PD-L1 were from CST, granzyme B conjugated to Alexa 647 and Pan-CK were from Abcam, CD4 and CD8 conjugated to Alexa 488 were from Novus Biological. IF buffers were from DAKO and blocking buffer was from Thermo Fisher. For in vivo experiments, anti-PD-1 (BE0146), anti-CD4 (L3T4) anti-CD8 (YTS169.4), and anti- CXCL9 (MIG-2F5.5) were from BioXCell as previously described [3]. Goat anti-mouse CXCL10 was given by Dr. Robert Strieter. Isotype control Ab was from Sigma. Tissue digestion buffer was from Miltenyi and used according to manufacturer’s instructions. T cell purification columns were from R&D. RNA isolation kit was from Qiagen.

### 2.2. LKR-13 and DC Cell Culture

LKR-13 lung cancer cell line and DC from femurs of syngeneic 129-E mice were propagated as previously described [3]. The cells were mycoplasma and murine viral pathogen-free. The LKR-13 cell line was used up to the 10th passage to avoid over passage [4]. For in vitro work, each time the cells are trypsinized is regarded as a passage. The cells reach the 10th passage after three weeks. DC were cultured for 7 days in GM-CSF (20 ng/mL) and IL-4 (4 ng/mL) culture media (CM) in flasks coated with 2% gelatin (Sigma). The cultured DC were used on day 7 for experiments. The DC expressed 70–90% of MHC class I and MHC class II (data not shown).

### 2.3. CCL21 Transduction and K-Ras Peptide Pulsing of DC

DC were transduced with CCL21 or control vector as previously described [3]. The transduced DC were pulsed with MHC Class I (CD8 restricted) and MHC Class II (CD4 restricted) K-Ras peptides (K-Ras MHC Class I [LVVVGADGV] and MHC Class II [MTEYKLVVVGADGVG] (New England Peptides) CCL21 transduced DC produced 10–16 ng CCL-21/10^6^ cells /24 h evaluated by CCL-21-specific ELISA. For peptide pulsing, DC cells (50 × 10^6^) were incubated with 10 μM MHC Class I and 10 μM MHC Class II tumor-specific K-Ras peptides for 3 h in 5 mL of RPMI. The cells were washed 3X with PBS and reconstituted in PBS for s.c. injection in the contralateral flank of the tumor in tumor-bearing mice. 5 × 10^6^ DC pulsed with tumor peptides/vaccination were utilized. DC pulsed with MHC Class I peptide (LVVVGADGV) and co-cultured with CD8 T cells or pulsed with MHC Class II peptide (MTEYKLVVVGADGVG) and co-cultured with CD4 at a ratio of 1:5 secreted 1500 pg/mL and 1355 pg/mL of IFNγ as determined by ELISA. Control peptide (FECNTAQAC) did not stimulate detectable levels of IFNγ.

### 2.4. CCL21 Transduction and Tumor Lysate Pulsing of DC

The transduced DC were pulsed with LKR-13 tumor lysates prepared by digesting LKR-13 tumors (day 15–20) and used as previously described [3].

### 2.5. LKR-13 Tumorigenesis in 129- E mice

Tumorigenesis was conducted to test if CCL21-DC peptide or tumor lysate vaccine combined with anti-PD-1 would augment therapy over monotherapy. LKR-13 tumor cells (10^6^) were injected s.c. and tumor volume calculated as previously described [3]. 12 days following tumor inoculation, mice with 150 mm^3^ tumors were injected i.p. with anti-PD-1-specific (200 µg/dose) or isotype IgG2b Ab (200 µg/dose) every 48 h for 3 weeks. CCL21- DC tumor peptide or lysate Ag pulsed vaccine (5 × 10^6^) was injected (1X/week X3) by s.c. administration on the left contralateral flank of the tumor. For sequential therapy CCL21-DC peptide or lysate vaccine was administered one week prior to combined therapy with CCL21-DC peptide or lysate vaccine plus anti-PD-1 to test if sequential administration of vaccine prior to combined therapy would lead to a better outcome of eradicating tumors in mice. In the combined therapy CCL21-DC tumor Ag vaccine was administered s.c. 1X/wk for three weeks and anti-PD-1 was administered 3x/wk via i.p. injections for three weeks. To test for immunological memory, mice were re-challenged with 10^6^ LKR-13 tumor cells as previously described [3].

### 2.6. Flow Cytometry of CD8 T Cells Expressing Perforin

To determine CD8 T cells expressing perforin in the TME following therapy, flow cytometry analyses were performed as previously described [3].

### 2.7. H&E of Tumor Microenvironment

H&E staining was conducted on tumor sections as previously described [5].

### 2.8. Immunohistochemistry (IHC) and Immune Fluorescence (IF) Staining of Tumor Microenvironment

IHC and IF staining was conducted on tumor sections as described previously [5]. IF images were acquired using Zeiss microscope (10 fields/slide X8 were scanned using 4X objective) and analyzed with MATLAB software (R 2011b, MathWorks).

### 2.9. CD4 T Cell, CD8 T Cell, CXCL9, and CXCL10 Depletion

Mice bearing 12-day tumors were treated with combined therapy and CD4 T cells and CD8 T cells were depleted as previously described [3]. To deplete CXCL9 or CXCL10, mice were treated with anti-CXCL9 or anti-CXCL10 (200 μg/dose) via i.p. injection every 48 h for the duration of the experiment. Anti-CXCL9 or anti-CXCL10 depleted the respective chemokine in the TME as determined by ELISA (data not shown).

### 2.10. RNA Sequencing of Tumor Microenvironment

RNA-sequencing was conducted to quantify cell scores and gene expression profiles in the TME. RNA sequencing and analyses were performed by MedGenome according to established protocols. Briefly, total RNA was isolated from the tumors by using RNA easy kit one week following the 2nd week of treatment. The RNA samples were sequenced on Illumina HiSeq platform with TruSeq Stranded mRNA Library Prep Kit. Bioinformatics analysis was performed by MedGenome. QC check, contamination removal and read alignment was performed to the reference mouse genome. The aligned reads were used for estimating expression of the genes using HTSeq-0.6.1 software. Hierarchical clustering analysis was performed for normalized counts. Euclidean distance and the complete linkage clustering method was used for hierarchical clustering. Analysis was performed using the R language and additional packages: ggplot2, reshape2 and ggrepel. Analysis of TME was performed with OncoPeptTUMETM. The TME sample was analyzed at multiple levels. First immune cell compartment was stratified into two different immune cell types; adaptive immune cells, which are CD8 T cells, CD4 T cells, Treg cells, B cells and innate immune cells, which are M1 and M2 macrophages. The signature-based scoring was calculated using the ssGSEA (single cell Gene Set Enrichment Analysis) method provided by R package ESTIMATE.

### 2.11. TIL In Vitro Cytolysis of LKR-13 Cells

TIL activity against LKR-13 cells was assayed as previously described [3].

### 2.12. Statistical Analyses

GraphPad Prism was used for statistical analysis as previously described [3]. All data are presented as mean ± SE. *p* values < 0.05 were considered significant.

### 2.13. Ethical Disclosure

All animal work was conducted in accord with the Veterans Affairs Institutional Animal care and Use Committee guidelines: id D16-00002. The Animal Care and Use Committee review board approved all the studies involving animals.

## 3. Results

Combined therapy significantly inhibited tumor growth in comparison to monotherapy and the effects of therapy were CD4 T and CD8 T cell dependent (Figure 1A,B,D,E). Depletion of CXCL9 and CXCL10 abrogated the anti-tumor activity of combined therapy (Figure 1C).

CCL21-DC peptide vaccine administered prior to combined therapy led to 100% tumor radication. Around 10% of the cured mice had a tumor recurrence 4 weeks following tumor eradication (Table 1).

However, the recurred tumors grew extremely slowly not exceeding at 25.6–86.6 mm^3^. The cured mice rejected a secondary tumor challenge and remained tumor-free (Table 1). Similarly, tumor lysate pulsed CCL21-DC vaccine administered prior to combined therapy led to 90% tumor eradication. Individual depletion of CD4 T cells, CD8 T cells (Figure 1A,B) and CXCL9, CXCL10 (Figure 1C), abrogated the anti-tumor activity of combined therapy. Mice that exhibited treatment-induced tumor eradication, rejected a secondary tumor challenge demonstrating long term immunological memory (Table 1). Combined therapy caused 21-fold tumor weight reduction in comparison to 3-fold by monotherapy (Figure 1D,E). DC peptide, DC tumor lysate, Control vector (CV)-DC peptide plus anti-PD-1 and CV-DC tumor lysate vaccine plus anti-PD-1 caused 10% tumor eradication (data not shown).

Figure 2A: There was enhanced CD8 T cell perforin expression following combined therapy. The experiment was repeated twice (*n* = 4 mice/group).

Figure 2B: TILs from mice treated with sequential therapy with vaccine prior to combined therapy had the greatest lytic activity. There was increased CD4 T cells (3-fold), CD8 T cells (10-fold), B cells (13-fold), IFN γ (11- fold), TNF α (2-fold), perforin (3-fold), granzyme B (2- fold), INOS (2-fold), Arginase (2-fold), CXCR3 (7- fold), CXCL9 (5-fold), CXCL10 (6-fold). DC were increased according to the following markers (IL3ra, Itgam, Anpep, Sell, CD68) in the CCL21-DC peptide vaccine plus anti-PD-1 treatment in comparison to control by RNA sequencing. There was increase expression of VISTA in the TME following combined therapy (Table 1).

Figure 2C–E: H&E, IHC and IF of tumor microenvironment demonstrated enhance immune infiltrates and reduced tumor burden following combined therapy. Red denotes cytokeratin in tumor and pink denotes perforin in activated CD8 T cells. This is a representative IF figure for the markers cytokeratin, PD-L1, CD4, CD8, perforin and granzyme B. Image analysis of IF staining of tumor sections revealed enhanced CD4 T cells, CD8 T cells, perforin and granzyme B but reduced tumor-specific cytokeratin following therapy (Figure 2E).

## 4. Discussion

In a previous study, we demonstrated that CCL21-DC Tumor Ag vaccine combined with PD-1 blockade leads to tumor eradication in the K-Ras G12Dp53 null model of lung cancer [3]. To determine the generality of our therapeutic approach, we evaluated the therapy in the K-Ras lung cancer model. CCL21-DC tumor antigen vaccine induced infiltration of T cells that was activated by PD-1 blockade. The individual therapies were not as effective as combined therapy. NSCLC patients who have low responses to PD-1 blockade have low T cell infiltrates [6]. PD1 antibody in clinical trials [7,8,9] have shown effectiveness in tumors with high mutation burden [10,11].

Furthermore, active tumor-induced immune suppression promoting tumor progression is through tumor associated macrophages (TAMs), Tregs and MDSC. Treg cells downregulate activation and expansion of anti-tumor reactive T cells [12,13,14] and NK cells [15,16]. MDSC have a strong suppressive activity on T cells and NK cells in cancer [17]. There were no significant changes in TAMs. However, Vista, Arg and iNOS expression were the highest in the combined treatment group. This suggests that MDSC activity may need to be inhibited for optimum tumor control in later stages of tumor growth.

We evaluated a tumor chemokine signature that identifies effector T cell recruitment into the TME and promotes effective cell-mediated anti-tumor activity. In accordance with the literature [16], CXCL9 or CXCL10 depletion abolished the anti-tumor efficacy of combined therapy.

## 5. Conclusions

Anti-PD-1 or CCL21-DC tumor Ag vaccine monotherapy reduced tumor burden without tumor eradication. Neutralization of CD4, CD8, CXCL9, and CXCL10 abrogated the anti-tumor activity of combined therapy. Eradication of tumors was enhanced to 90% by administering CCL21-DC tumor Ag vaccine prior to combined therapy. Targeting inhibitory immune checkpoint molecules in combination with therapeutic cancer vaccines have the potential for long-term cancer-free survival.

## Figures and Tables

**Figure 1 vaccines-08-00655-f001:**
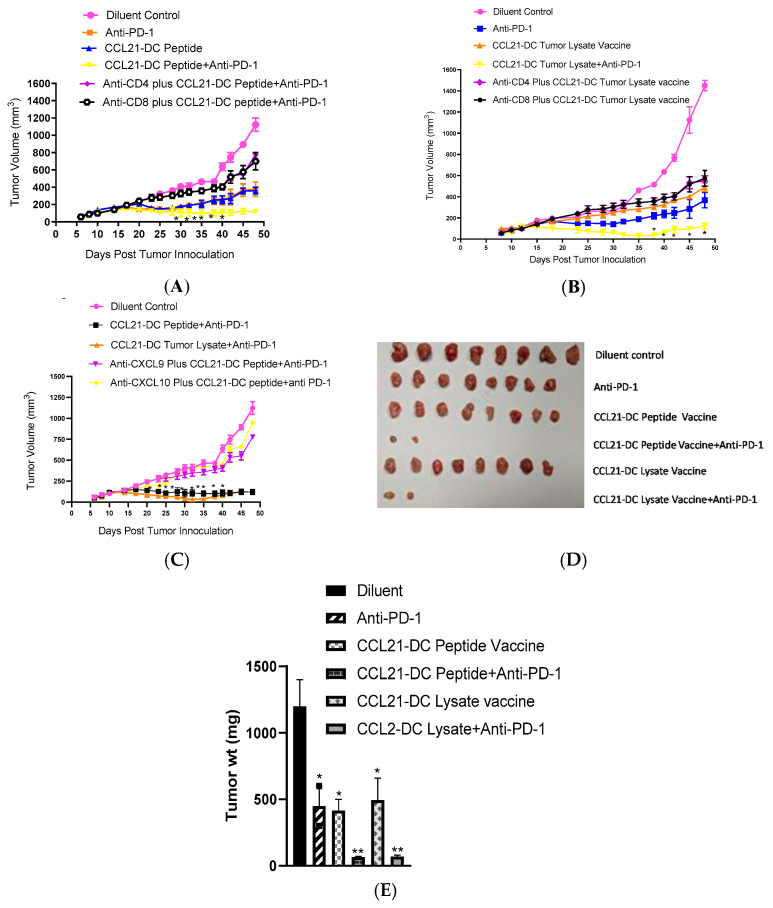
Combined therapy significantly inhibited tumor growth in comparison to monotherapy and the effects of therapy were CD4 T and CD8 T cell dependent (**A**,**B**,**D**,**E**). Depletion of CXCL9 and CXCL10 abrogated the anti-tumor activity of combined therapy (**C**).

**Figure 2 vaccines-08-00655-f002:**
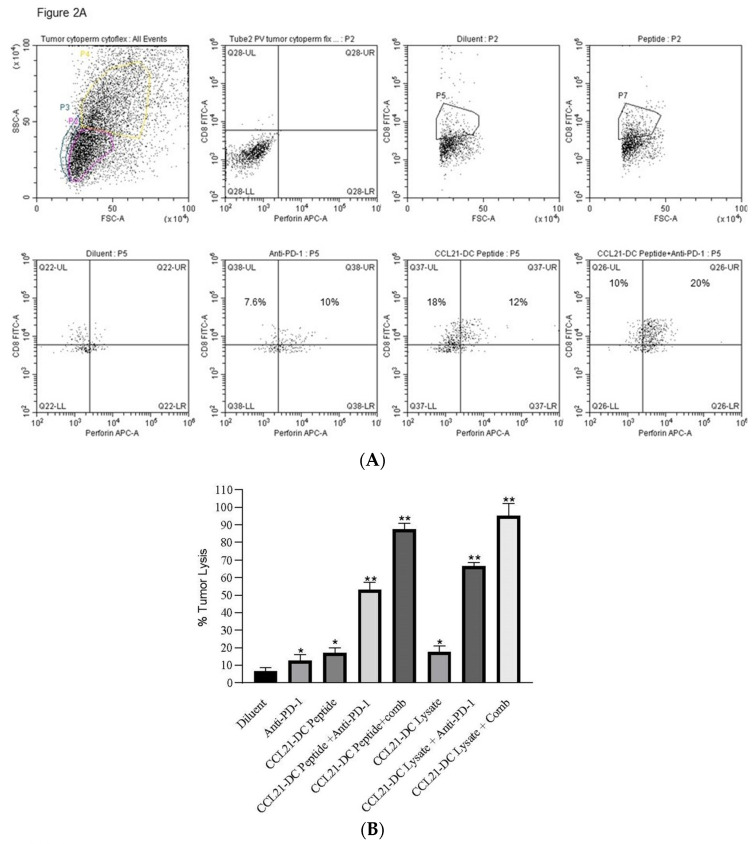
Combined Therapy Enhances CD8 T cells Expressing Perforin, TIL Cytolysis Immune Infiltrates and Reduced Tumor Burden (**A**–**E**).

**Table 1 vaccines-08-00655-t001:** ND = Not determined Table 1 LKR-13 cells (10^6^) were inoculated by s.c. injection in to 129-E strain of mice. 12-days following tumor inoculation, mice were treated with: (1) Diluent control, (2) Anti-PD-1, (3) CCL21-DC peptide vaccine, (4) CCL21-DC peptide vaccine + anti-PD-1, (5) CCL 21-DC peptide vaccine prior to combination (6) CCL21-DC lysate vaccine, (7) CCL21-DC lysate vaccine + anti-PD-1 and (8) CCL21-DC lysate vaccine prior to combination (9) Anti-CD4 plus CCL21-DC peptide vaccine + anti-PD-1 and (10) Anti-CD8 plus CCL21-DC peptide vaccine + anti-PD-1 (11) Anti-CD4 plus CCL21-DC lysate vaccine + anti-PD-1 and (12) Anti-CD8 plus CCL21-DC lysate vaccine + anti-PD-1 1X/wk for 3 weeks. Combined therapy eradicated 75% of the tumors. Sequential therapy with vaccine prior to combined therapy is most effective causing 90% tumor eradication. Table is representative of the number of mice that eradicated their tumors following therapy. Results are representative of an independent experiment. The experiment was repeated twice (*n* = 10–12 mice/group).

Treatment Groups	#Of Mice That Eradicated Tumors	#Of Mice That Rejected Re-Challenge
Dil. Control	0/12	ND
Anti- PD-1	0/12	ND
CCL21- DC Peptide vaccine	0/12	ND
CCL21-DC Peptide vaccine + anti- PD-1	9/12	9/9
CCL21-DC Peptide vaccine prior to combined therapy	9/10	9/9
CCL21- DC Lysate vaccine	0/12	ND
CCL21-DC Lysate vaccine + anti- PD-1	9/12	9/9
CCL21-DC Lysate Vaccine prior to combined therapy	9/10	9/9
Anti-CD4 + CCL21-DC Peptide vaccine + anti- PD-1	0/10	ND
Anti-CD8 + CCL21-DC Peptide vaccine + anti- PD-1	0/10	ND
Anti-CD4 + CCL21-DC Lysate vaccine + anti- PD-1	0/10	ND
Anti-CD8 + CCL21-DC Lysate vaccine + anti- PD-1	0/10	ND

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
