# Peer review of "Immune Checkpoint Blockade Enhances Immune Activity of Therapeutic Lung Cancer Vaccine"

_vaccines, 2020, doi:10.3390/vaccines8040655_

Round 1

Reviewer 1 Report

The manuscript presented by Kadam et al. presents a set of data regarding immunotherapy in the context of experimental lung cancer vaccine. The theme is suitable for publication in this journal, but the manuscript presents some technical problems, namely:

- Running title: I think the term "eradication" is not appropriate in this session;

- Abstract: I think the abstract is not written properly. The text is not attractive to readers because it has many abbreviations and is written in a descriptive way. I think that this session deserves adjustments and better text formatting;

- Keywords: Are the authors unable to limit the manuscript to 5-6 keywords?

- Introduction: This section is too short and has many abbreviations, making it difficult to read and understand the text. I suggest that the introduction be reviewed and properly presented; 

- Material and methods: This section needs to be revised and formatted. For example: H&E, bold text, etc. I think this session deserves a lot of writing and formatting by the authors;

- Results: graphs and tables are presented in a confusing way. authors need to report data with support for graphs and tables. The graphics are not graphic quality, the figures are not very informative. It is a session that is difficult to read and understand;

- I think the topics "future perspective" and "executive summary" should be removed from the manuscript.

Author Response

Thank you for your thoughtful constructive feedback, they are much appreciated and all noted.

We provide a point by point response to your review below:

Comments of Reviewer 1

 Comment 1: Running title: I think the term "eradication" is not appropriate in this session;

Response to comment 1: We have removed “ eradication” from the running title.

Comment 2:  Abstract: I think the abstract is not written properly. The text is not attractive to readers because it has many abbreviations and is written in a descriptive way. I think that this session deserves adjustments and better text formatting.

Response to comment 2: We have removed all abbreviations from the abstract.

Comment 3: - Keywords: Are the authors unable to limit the manuscript to 5-6 keywords?

Response to comment 3: We have reduced the key words to 6.

Comment 4: Introduction: This section is too short and has many abbreviations, making it difficult to read and understand the text. I suggest that the introduction be reviewed and properly presented;

Response to comment 4: We have removed the abbreviations from the introduction.

Comment 5:  Material and methods: This section needs to be revised and formatted. For example: H&E, bold text, etc. I think this session deserves a lot of writing and formatting by the authors;

Response to comment 5: We have removed the bold text from method section of H&E section. We have revised  the materials and method section.

Comment 6: Results: graphs and tables are presented in a confusing way. authors need to report data with support for graphs and tables. The graphics are not graphic quality, the figures are not very informative. It is a session that is difficult to read and understand;

Response to comment 6: Graphs and tables are referred in the text according to journal style.

Comment 7:  I think the topics "future perspective" and "executive summary" should be removed from the manuscript.

Response to Comment 7: We have removed future perspective and executive summary from the manuscript.

Reviewer 2 Report

Immune Checkpoint Blockade Enhances Immune Activity of Therapeutic Lung Cancer Vaccine

P. Kadam et al

The current study evaluates if PD-1 blockade therapy along with CCL21-DC tumor Ag vaccine would be beneficial to patients with lung cancer. Obviously a great amount of work went into this study.

There are several things, however, that keep this from being publication ready.

Introduction

The introduction is replete with abbreviations. This area needs to be cleaned up in order for the audience to read it easily.

Materials and methods

It is stated that the LKR-13 cell line was used up to the 10th passage.  Can the authors provide the amount of time to reach the 10th passage? I am concerned about over-passage (Hughes P, Marshall D, Reid Y, Parkes H, Gelber C. The costs of using unauthenticated, over-passaged cell lines: how much more data do we need? Biotechniques. 2007;43(5):. doi:10.2144/000112598). Is it common to use this cell line up to the 10th passage?  Please provide information about this issue.

Provide complete information concerning the H&E and Immunohistochemistry (IHC) and Immune Fluorescence (IF) Staining techniques. The details for these techniques are missing.

Minor point: Why is the CD4T-cell, CD8T-cell, CXCL9 and CXCL10 depletion in bold?

Results

The tables and figures are not presented in a consistent fashion and this made the manuscript very confusing to read. Figure and table legends sometimes are included in the text and also at the end of the document. Sometimes the figure legends only describe one part, for example figure 2B. The last three rows of table 2 do not line up with the preceding rows. The photomicrographs need scale bars on at least one of the photos. Things like 20X really provide no information since objectives on cameras are different from microscope to microscope. Also, arrows or arrowheads should be used on the photomicrographs in order to point out the differences the authors would like to make.

Author Response

Comments of Reviewer 2

Thank you for your thoughtful constructive feedback, they are much appreciated and all noted.

We provide a point by point response to your review below:

 Comment1: The introduction is replete with abbreviations. This area needs to be cleaned up in order for the audience to read it easily.

 Response to comment 1: We have removed abbreviations from the introduction.

Comment 2: Materials and methods: It is stated that the LKR-13 cell line was used up to the 10th passage. Can the authors provide the amount of time to reach the 10th passage? I am concerned about over-passage (Hughes P, Marshall D, Reid Y, Parkes H, Gelber C. The costs of using unauthenticated,

over-passaged cell lines: how much more data do we need? Biotechniques. 2007;43(5):. doi:10.2144/000112598). Is it common to use this cell line up to the 10th passage? Please provide information about this issue.

Response to comment 2: For in-vitro work each time the cells are trypsinzed is regarded as passage. The cells reach the 10th passage after three weeks.

Comment 3: Provide complete information concerning the H&E and Immunohistochemistry (IHC) and Immune Fluorescence (IF) Staining techniques. The details for these techniques are missing.

Response to comment 3: We have included references describing H&E, IHC and IF.

Comment 4: Minor point: Why is the CD4T-cell, CD8T-cell, CXCL9 and CXCL10 depletion in bold?

 Response comment 4: We have removed the bold font.

Comment 5: Results: The tables and figures are not presented in a consistent fashion and this made the manuscript very confusing to read. Figure and table legends sometimes are included in the text and also at the end of the document.

Response to comment 5: All figures and tables are referred in the text. Figure legends appear according to journal style.

Comment 6: Sometimes the figure legends only describe one part, for example figure 2B. The last three rows of table 2 do not line up with the preceding

rows. The photomicrographs need scale bars on at least one of the photos. Things like 20X really provide no information since objectives on cameras are different from microscope to microscope. Also, arrows or arrowheads should be used on the photomicrographs in order to point out the differences the authors would like to make.

Response to comment 6: We have fixed the legends for the figure 2B and aligned the rows of Table 1 and 2. In the photomicrographs the color of the stain described the targets. All figures have been revised for better quality.

Round 2

Reviewer 1 Report

The manuscript reviewed by the authors is adequate to be appreciated by the editors.

Author Response

Response to reviewer comment

 Reviewer 1.

We thank the reviewer for the comments that has bolstered the presentation of our manuscript. Please find a point- by- point response to the reviewer comments.

 Comment 1. English language and style are fine/minor spell check required. 
 Response 1. The manuscript has been edited for English language and spell check.

 Comment 2. Does the introduction provide sufficient background and include all relevant references? Are the methods adequately described? Are the results clearly presented?

Response 2. The introduction, methods and results are revised for clarity with all relevant references.

Comment 3. Are the conclusions supported by the results?

Response 3. Conclusion has been added that is supportive of the results.

Reviewer 2 Report

vaccines-93280 -v2

Immune Checkpoint Blockade Enhances Immune Activity of Therapeutic Lung Cancer Vaccine

Running title: Anti-PD-1 plus Tumor Vaccine Attenuates Cancer

Kadam et al

This is an improved manuscript and I am satisfied, for the most part, concerning what the authors have revised. There are two items that still need to be addressed.

Materials and methods

It would still be useful to understand the amount of time to reach the 10th passage for the LKR-13 cell line. Can the authors address the issue of over-passage? (Hughes P, Marshall D, Reid Y, Parkes H, Gelber C. The costs of using unauthenticated, over-passaged cell lines: how much more data do we need? Biotechniques. 2007;43(5):. doi:10.2144/000112598).

Results

Put a scale bar on the first 20 X and 60 X photomicrograph for figure 2C. Put a scale bar on the first photomicrograph of figure 2B.

Author Response

Reviewer 2

We thank the reviewer for the comments that has bolstered the presentation of our manuscript. Please find a point- by- point response to the reviewer comments.

 Comment 1. Are the methods adequately described?

 Response 1. We have edited the methods for adequacy.

 Comment 2. Are the results clearly presented? Are the conclusions supported by the results?

 Response 2. The results have been edited for clear presentation and conclusion added that is supportive of the results.

 Comment 3. Materials and methods; It would still be useful to understand the amount of time to reach the 10th passage for the LKR-13 cell line. Can the authors address the issue of over-passage? (Hughes P, Marshall D, Reid Y, Parkes H, Gelber C. The costs of using unauthenticated, over-passaged cell lines: how much more data do we need? Biotechniques. 2007;43(5):. doi:10.2144/000112598).

Response 3.  We have added in the manuscript that the LKR-13 cell line was used up to the 10th passage to avoid over passage [4].  For in-vitro work each time the cells are trypsinzed is regarded as a passage. The cells reach the 10th passage after three weeks. The suggested reference (4) has been added.

Comment 4. Results. Put a scale bar on the first 20 X and 60 X photomicrograph for figure 2C. Put a scale bar on the first photomicrograph of figure 2D.

Response 4. We have placed scale bars in figure 2C and 2D.
